# CONSTRUCTIVE LARGE LANGUAGE MODELS ALIGNMENT WITH DIVERSE FEEDBACK

## ABSTRACT

In recent research on large language models (LLMs), there has been a growing emphasis on aligning these models with human values to reduce the impact of harmful content. However, current alignment methods often rely solely on singular forms of human feedback, such as preferences, annotated labels, or natural language critiques, overlooking the potential advantages of combining these feedback types. This limitation leads to suboptimal performance, even when ample training data is available. In this paper, we introduce Constructive and Diverse Feedback (CDF) as a novel method to enhance LLM alignment, inspired by constructivist learning theory. Our approach involves collecting three distinct types of feedback tailored to problems of varying difficulty levels within the training dataset. Specifically, we exploit critique feedback for easy problems, refinement feedback for medium problems, and preference feedback for hard problems. By training our model with this diversified feedback, we achieve enhanced alignment performance while using less training data. To assess the effectiveness of CDF, we evaluate it against previous methods in three downstream tasks: question answering, dialog generation, and text summarization. Experimental results demonstrate that CDF achieves superior performance even with a smaller training dataset.[1]

## 1 INTRODUCTION

In recent years, large language models (LLMs) have demonstrated remarkable proficiency in fulfilling diverse information requirements (Chowdhery et al., 2022; Bubeck et al., 2023; Touvron et al., 2023; Li et al., 2023; Muennighoff et al., 2023). An emerging research emphasis in this domain involves *aligning LLMs with human values* to mitigate the risk of generating misleading or harmful content (Bai et al., 2022a; Wang et al., 2023c). A representative method for achieving this alignment is Reinforcement Learning from Human Feedback (RLHF) (Christiano et al., 2017). This technique trains the model's decision-making policy using a reward model derived from human preferences, which is widely employed in various studies (Ziegler et al., 2019; Stiennon et al., 2020; Nakano et al., 2021; Ouyang et al., 2022; Bai et al., 2022a; Akyürek et al., 2023; Wu et al., 2023).

Nevertheless, a notable drawback of current alignment techniques is their considerable reliance on a large amount of human-annotated preference data (Casper et al., 2023). For instance, Ouyang et al. (2022) reported the necessity of millions of interactions with human annotators to achieve satisfactory performance levels. In task-specific scenarios such as summarization, approximately 100,000 specialized data points remain indispensable (Stiennon et al., 2020). This data-intensive demand can be attributed to the predominant utilization of a single form of feedback, overlooking the potential advantages of combining these feedback types.

We establish an analogy between the alignment process and students' educational experiences. Similar to students facing diverse problems of varying topics and difficulty levels in their coursework, alignment efforts encounter a range of distinct tasks. With regard to the constructivist learning theory (Vygotsky & Cole, 1978), the "Zone of Proximal Development" (ZPD) principle categorizes tasks into three types based on their level of difficulty: (i) tasks that can be independently solved, (ii) tasks that can be solved with assistance, and (iii) tasks that remain unsolvable even with assistance. To effectively address these diverse challenges, students often adjust their approach based on

---

[1]We will release our code for reproducibility upon acceptance.

the task's nature and difficulty, seeking assistance or guidance when needed. Specifically, for tasks that can be independently solved, students may arrive at correct solutions, but concise guidance can foster deeper comprehension and enhanced understanding. Tasks requiring assistance demand more refined responses and can benefit from iterative feedback, as nuanced suggestions may be less straightforward to internalize. For tasks deemed unsolvable, providing direct advice on incorrect answers may prove ineffective. In such cases, learning through preferences among potential solutions emerges as a more efficient approach, particularly for complex problems that surpass the student's capability.

We postulate that this analogy and its corresponding learning principle are highly relevant to the alignment of LLMs. Achieving alignment with LLMs can be enhanced by obtaining diverse feedback, which is tailored to match the difficulty level of each problem. When dealing with simpler tasks that LLMs can handle independently, we recommend the use of critique feedback, mirroring the acquisition of natural language critiques. As tasks become more complex and require assistance, refinement feedback becomes invaluable, enabling LLMs to learn from the improvements suggested in their responses. For tasks that remain unresolvable even with assistance, we advocate for preference-based learning, involving the incorporation of preference feedback when multiple potential answers are available.

Inspired by the constructivist learning theory, we present Constructive and Diverse Feedback (CDF), a novel method designed to enhance the alignment process by integrating diverse feedback mechanisms. Our CDF method employs a combination of critique, refinement, and preference feedback to efficiently tackle tasks of varying difficulty levels, thereby enhancing the alignment of LLMs with human values. We begin with an unaligned base model and initially generate model outputs for a predetermined set of problems. We employ the perplexity of these outputs as our metric to assess the difficulty level of the problems. Subsequently, we categorize the problems into three difficulty levels: *easy*, *medium* and *hard*. For *easy* problems, we gather critique feedback; for *medium* ones, we collect reference feedback; for most challenging *hard* problems, we solicit preference feedback.

The main contributions of our work are three-fold:

1. We present CDF as an innovative method designed to enhance the alignment process by integrating diverse feedback mechanisms according to the ZPD principle.

2. We propose a differentiated feedback approach that employs and combines various types of feedback tailored to the varying difficulty levels of the problems.

3. We present experimental results across three downstream tasks, demonstrating that our CDF method achieves superior performance even with a smaller training dataset. Additional experiments and analyses reinforce the effectiveness of CDF.

## 2 RELATED WORK

### 2.1 RESEARCH ON THE ALIGNMENT OF LLMS

In recent years, the task of fine-tuning language models to align with human values has gained paramount importance, driven by the imperative to reduce the generation of incorrect, misleading, or harmful content in dialog completions (Bai et al., 2022a; Ouyang et al., 2022; Liu et al., 2023b; Wang et al., 2023c). Reinforcement learning (RL) has become the predominant technique in numerous prior studies tackling this challenge. RL frames the generation process as a Markov decision process and optimizes the policy model to maximize a proxy reward, establishing itself as a pivotal method in this context. For instance, Ziegler et al. (2019) were pioneers in investigating the RLHF method for stylistic continuation and summary generation. Bai et al. (2022b) introduced the concept of LLM alignment along with the HHH (helpful, harmless, honest) principle, applying RLHF to achieve alignment. Ouyang et al. (2022) introduced InstructGPT, which was subsequently applied to the renowned ChatGPT. In addition to RLHF, alternative training methods have been explored. Liu et al. (2023a) proposed CoH, which learned from both good and bad responses. Rafailov et al. (2023) introduced the DPO algorithm to mitigate the instability of PPO training, derived from the classic Bradley-Terry model of reward proxy learning. Song et al. (2023) extended DPO to scenarios where a prompt can elicit more than two possible responses with annotated human preference order.

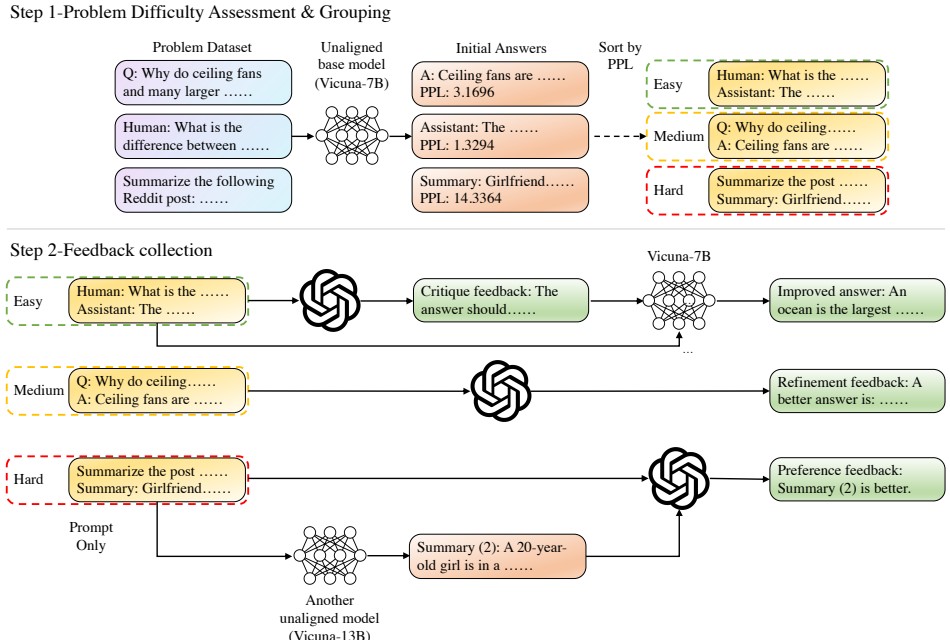

Figure 1: The illustration of our method CDF, describing the procedure we collect feedback for problem dataset $\mathcal{D}$.

## 2.2 LEARNING FROM VARIOUS TYPES OF FEEDBACK

Existing literature has explored diverse forms of feedback to enhance model predictive capabilities. These methods can be classified as (i) preferences that involve pairwise comparisons or rankings (Ouyang et al., 2022; Bai et al., 2022a; Gao et al., 2022; Zhu et al., 2023); (ii) natural language critiques (Tandon et al., 2021; Schick et al., 2022; Scheurer et al., 2022; Saunders et al., 2022; Madaan et al., 2023); and (iii) direct textual refinements of generated outputs (Shi et al., 2022; Welleck et al., 2022). Saunders et al. (2022) introduced a learning paradigm known as Self-Critique, where a model evaluated its own outputs in natural language and then refined itself based on these critiques. Bai et al. (2022b) extended the above idea through Reinforcement Learning from AI Feedback (RLAIF), initially training a model using Self-Critique and subsequently using that model as the source of preference feedback for RLHF. Several studies, such as Schick et al. (2022) and Wang et al. (2023b), have gathered natural language feedback and corresponding refinements from online forums and Wikipedia. Specialized feedback types have also been developed for task-specific applications. For instance, Gao et al. (2022) employed accuracy metrics in extractive question-answering as feedback for policy fine-tuning, while Uesato et al. (2022) used the correctness of both the solution process and the final outcomes as feedback.

Unlike prior methods, we present CDF to align LLMs with human values through a differentiated feedback mechanism. CDF distinguishes itself by incorporating three distinct types of feedback, each calibrated to the problem's difficulty level, enhancing the effectiveness of model training.

## 3 METHOD

In this section, we provide a detailed explanation of our proposed CDF method, which is depicted in Figure 1.

## 3.1 THE CONSTRUCTION OF PROBLEM DATASET

To commence our study, we assemble a problem dataset, denoted as $\mathcal{D}$, for the purposes of feedback collection and model training. Specifically, we consider three specific datasets for our three distinct tasks within the realm of natural language processing: the *WebGPT-comparison* dataset for question answering, the *HH-RLHF* dataset for dialogue generation, and the *OpenAI-Summarize-TLDR* dataset for text summarization. We randomly select 10,000 samples from each of these datasets and combine them to create a comprehensive problem dataset $\mathcal{D}$, which forms the experimental foundation of our study.

## 3.2 DIFFICULTY CALCULATION AND GROUPING

Initially, in our experiment, we use the Vicuna-7B model as our base model. To evaluate the difficulty level of each problem in the dataset $\mathcal{D}$, we first generate an answer $a$ for each problem $q$ in $\mathcal{D}$ using a greedy search algorithm. Then, we calculate the perplexity $PPL(a, q)$ for each answer $a$ using the following formula:

$$PPL(a,q) = \left( \prod_{i=1}^{l} p(a_i | a_{<i}, q) \right)^{\frac{1}{l}} = \exp \left( \frac{1}{l} \sum_{i=1}^{l} \log p(a_i | a_{<i}, q) \right)$$

where $l$ represents the length of the answer $a$. We adopt perplexity as the metric for evaluating problem difficulty since it is a representative measure of semantic uncertainty. This concept has been widely utilized in previous studies that address uncertainty in language generation tasks. Similar to how humans may express uncertainty when dealing with challenging questions during exams, a high perplexity score typically indicates that the model is also experiencing a hard question with a high degree of uncertainty while generating its answer.

**Grouping Strategy.** After computing the perplexity score for each problem in $\mathcal{D}$, we first organize the problems in ascending order of perplexity. We then evenly distribute them into three groups: the *easy* group $\mathcal{E}$ comprising problems with the lowest perplexity; the *medium* group $\mathcal{M}$ containing problems with moderate perplexity; and the *hard* group $\mathcal{H}$ including problems with the highest perplexity scores.

## 3.3 FEEDBACK COLLECTION

In the next stage of our experiment, we concentrate on collecting diverse feedback for the answers generated by the model. Concretely, we obtain the Critique feedback for problems in the *easy* group $\mathcal{E}$, the Refinement feedback for the *medium* group $\mathcal{M}$, and the Preference feedback for the *hard* group $\mathcal{H}$. Given the financial and logistical challenges linked to human annotation, we choose to employ the feedback generated by the GPT-3.5-turbo API as a substitute for human feedback. For a given problem $q$ and its corresponding answer $a$ in our dataset, we outline the feedback collection methods as follows:

- Critique (**Critic**): This feedback type provides constructive suggestions for enhancing answer $a$, enabling our base model to further refine itself. We solicit the API to provide improvement suggestions $s$ for answer $a$. Subsequently, the base model generates an improved answer $a_c$ based on $q$, $a$, and $s$.

- Refinement (**Refine**): In this feedback category, we receive an improved version of the original answer $a$, which is generated directly based on the problem $q$. We make an API query to enhance the answer $a$, resulting in the refined version $a_r$.

- Preference (**Prefer**): In this approach, the model selects the superior answer among two distinct answers generated for the same prompt. To this end, we utilize another unaligned model, Vicuna-13B, to produce an alternative answer $a'$ for the problem $q$. Subsequently, we employ the API to determine whether $a$ or $a'$ is the better option.

For additional clarification, we provide examples of prompts and API feedback collection requests in Appendix A.

## 3.4 TRAINING

The final phase of our approach involves model training using the amassed feedback dataset. To demonstrate the effectiveness of our proposed method, we conduct experiments by employing two distinct training methods: Reinforcement Learning from Human Feedback (RLHF) and Direct Preference Optimization (DPO), which are denoted as CDF-RLHF and CDF-DPP, respectively.

### 3.4.1 RLHF TRAINING

In the RLHF training context, we follow the second and third steps of the standard RLHF training procedure. Initially, we train a reward model using the gathered feedback. Subsequently, we fine-tune the policy using the Proximal Policy Optimization (PPO) algorithm.

**Comparison Dataset Construction.** To train a reward model, we initially convert the collected feedback into a comparison format, following Ouyang et al. (2022). This involves the following formats for different types of feedback: (i) for preference feedback, we use the answer pair $(a, a')$, where the API designates the preferred answer; (ii) for refinement feedback, we adopt the answer pair $(a_r, a)$, where $a_r$ is the improved answer indicated by the API; (iii), for critique feedback, we use the answer pair $(a_c, a)$, where $a_c$ represents the preferred version.

**Reward Model Training.** Before training the reward model, we partition the feedback dataset into a training set including 90% of the samples and a validation set comprising the remaining 10%. Then, we proceed to train the reward model for five epochs, starting from the base model. The model checkpoint with the highest validation accuracy is chosen for the subsequent PPO training phase.

**Policy Model Training.** In a manner similar to how we curated the dataset $\mathcal{D}$, we collect an additional 30,000 prompts that are distinct from $\mathcal{D}$ to train the policy model. Due to the limited data available in the *WebGPT-comparison* dataset, we also use alternative sources such as *eli5*, *trivia-qa*, and *ARC* for question-answering prompts. The base model is then trained on these prompts for one epoch using the PPO algorithm. To address concerns related to overfitting, we employ the PPO-ptx strategy Ouyang et al. (2022). The overall training objective can be formally expressed as:

$$\mathcal{L}_{RLHF} = \mathbb{E}_{(q,a)\sim\mathcal{D}^{ppo}} \left[ r_\theta(q, a) - \beta \log \left( \frac{\pi_{ppo}(a|q)}{\pi_{base}(a|q)} \right) \right] + \gamma \mathbb{E}_{x\sim\mathcal{D}^{ptx}} \left[ \log \pi_{ppo}(x) \right]$$

In this equation, $\mathcal{D}^{ppo}$ represents the dataset of collected prompts, $\mathcal{D}^{ptx}$ signifies the pretraining distribution, $\pi_{ppo}$ represents the learned PPO policy, and $\pi_{base}$ refers to the base model.

### 3.4.2 DPO TRAINING

We also train our base model using the DPO algorithm (Rafailov et al., 2023), as DPO offers improved training stability for optimizing the alignment target. DPO is trained on the training split of the comparison dataset, and the overall training objective can be formally expressed as:

$$\mathcal{L}_{DPO} = \mathbb{E}_{(q,a_w,a_l)\sim\mathcal{D}} \left[ \log \sigma \left( \beta \left( \log \frac{\pi_{dpo}(a_w|q)}{\pi_{base}(a_w|q)} - \log \frac{\pi_{dpo}(a_l|q)}{\pi_{base}(a_l|q)} \right) \right) \right] + \gamma \mathbb{E}_{x\sim\mathcal{D}^{ptx}} \left[ \log \pi_{dpo}(x) \right]$$

where $(a_w, a_l)$ denotes the answer pair of the problem $q \in \mathcal{D}$ and $a_w$ is the preferred one.

## 4 EXPERIMENTS

### 4.1 EVALUATION SETTINGS

We conduct experiments that specifically target three downstream tasks: question answering (QA), dialogue generation (Dial.), and text summarization (Summ.). To assess the effectiveness of our alignment strategy, we construct a separate test set comprising 300 prompts. This test set includes 100 prompts sampled for each of the three tasks and is entirely distinct from those used in both $\mathcal{D}$ and $\mathcal{D}^{ppo}$. We compare the performance of CDF against three benchmarks: the golden response annotated in the original dataset, our base model (Vicuna-7B), and a state-of-the-art model (LLaMa-2-7B-Chat). To further demonstrate the benefits of integrating multiple feedback types, we also

| Model | GPT-4 Scoring | | | | RM Scoring | | | |
|---|---|---|---|---|---|---|---|---|
| | **Avg.** | Summ. | QA | Dial. | **Avg.** | Summ. | QA. | Dial. |
| Golden | **2.83** | 2.34 | 3.34 | 2.80 | **-0.492** | -0.643 | -0.352 | -0.481 |
| Vicuna-7B | **3.49** | 2.94 | 3.81 | 3.72 | **0.609** | 0.641 | 0.698 | 0.488 |
| LLaMa2-7B-Chat | **3.65** | 2.90 | 3.99 | 4.05 | **0.814** | 0.654 | 0.961 | 0.828 |
| $RLHF_{Critic}$ | **3.77** | 3.26 | 4.18 | 3.88 | **0.941** | 0.995 | 1.101 | 0.726 |
| $RLHF_{Refine}$ | **3.80** | 3.28 | 4.15 | 3.98 | **0.951** | 0.981 | 1.115 | 0.756 |
| $RLHF_{Prefer}$ | **3.82** | 3.29 | 4.19 | 3.99 | **0.951** | 0.978 | 1.127 | 0.747 |
| CDF-RLHF (Ours) | **4.10** | 4.10 | 4.22 | 3.99 | **1.112** | 1.427 | 1.126 | 0.783 |
| $DPO_{Critic}$ | **3.77** | 2.65 | 4.43 | 4.24 | **0.896** | 0.349 | 1.235 | 1.105 |
| $DPO_{Refine}$ | **3.80** | 3.28 | 4.15 | 3.98 | **0.915** | 0.462 | 1.276 | 1.008 |
| $DPO_{Prefer}$ | **3.68** | 2.40 | 4.32 | 4.31 | **0.897** | 0.457 | 1.179 | 1.054 |
| CDF-DPO (Ours) | **4.37** | 4.24 | 4.52 | 4.36 | **1.389** | 1.523 | 1.437 | 1.207 |

Table 1: Evaluation results of CDF and baselines trained with RLHF and DPO algorithms in terms of GPT-4 and RM evaluation.

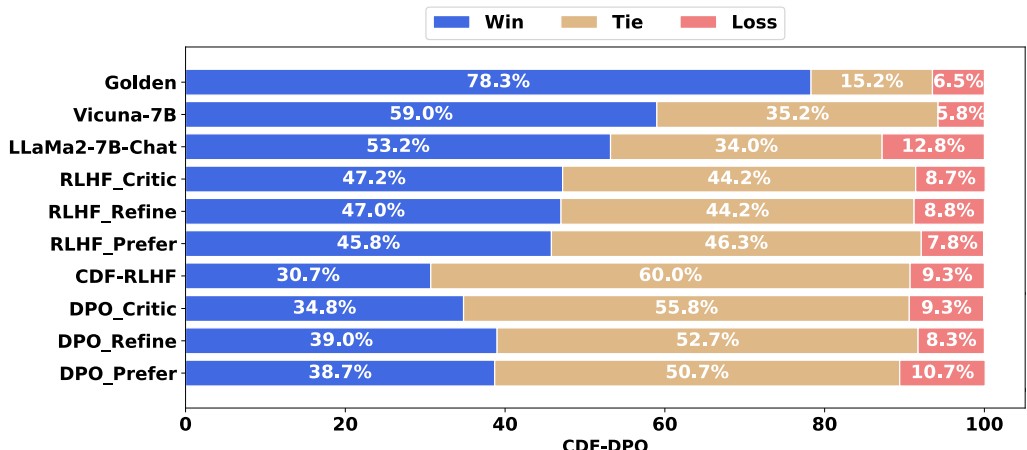

Figure 2: A comparative win rate plot for CDF-DPO against other methods via GPT-4 evaluation.

conduct RLHF and DPO training using the datasets restricted to single types of feedback, which are denoted as ($RLHF_{Prefer}$, $RLHF_{Refine}$, $RLHF_{Critic}$) and ($DPO_{Prefer}$, $DPO_{Refine}$, $DPO_{Critic}$), respectively. To accomplish this, we gather critique, refinement, and preference feedback for all problem-answer pairs in $\mathcal{D}$.

## 4.2 EVALUATION METRICS

We present our experimental results using three evaluation metrics: automatic assessment, model-based evaluation, and human-based evaluation. Our primary metric is based on an open-source reward model called OpenAssist[2], which automatically evaluates the quality of the generated content. We denote this metric as **RM** evaluation. In addition, recent studies have demonstrated the effectiveness of GPT-4 in evaluating chat assistant responses and aligning with human preferences (Zheng et al., 2023; Wang et al., 2023a). Therefore, we incorporate GPT-4 to rate the generated content on a scale from 1 to 5, where higher scores indicate better alignment with human values. We denote this metric as **GPT-4** evaluation. Finally, we acknowledge that human judgment serves as the gold standard for assessing alignment with human values. To address this, we engage human evaluators who perform pairwise comparisons among the top-performing models identified in automated evaluations.

---

[2]https://huggingface.co/OpenAssistant/oasst-rm-2-pythia-6.9b-epoch-1

| | | Ours Win (%) | Tie (%) | Ours Lose (%) | Gap (%) |
|---|---|---|---|---|---|
| CDF-RLHF vs Golden | Dial. | 70.0 | 15.6 | 14.4 | +55.6 |
| | QA. | 75.8 | 13.7 | 10.5 | +65.3 |
| | Summ. | 74.4 | 12.2 | 13.3 | +61.1 |
| | Avg. | 73.5 | 13.8 | 12.7 | +60.8 |
| CDF-RLHF vs RLHF$_{Prefer}$ | Dial. | 13.0 | 78.0 | 9.0 | +4.0 |
| | QA. | 25.0 | 59.0 | 16.0 | +9.0 |
| | Summ. | 53.3 | 23.3 | 23.3 | +30.0 |
| | Avg. | 29.7 | 54.5 | 15.9 | +13.8 |
| CDF-DPO vs Golden | Dial. | 80.0 | 11.1 | 8.9 | +71.1 |
| | QA. | 76.8 | 10.5 | 12.6 | +64.2 |
| | Summ. | 77.8 | 8.9 | 13.3 | +64.5 |
| | Avg. | 78.2 | 10.2 | 11.6 | +66.6 |
| CDF-DPO vs DPO$_{Refine}$ | Dial. | 32.2 | 51.1 | 16.7 | +15.5 |
| | QA. | 38.9 | 38.9 | 22.1 | +16.8 |
| | Summ. | 77.8 | 17.8 | 4.4 | +73.4 |
| | Avg. | 49.5 | 36.0 | 14.5 | +35.0 |

Table 2: Human evaluation results. This table shows the performance of CDF-RLHF and CDF-DPO against their counterparts across different contexts, showcasing Win, Tie, Lose, and Gap percentages. Here, Gap stands for the difference between Win and Lose percentages.

## 4.3 MAIN RESULTS

Table 1 offers a comprehensive summary of performance metrics for our CDF method and the compared benchmarks. We use OpenAssist and GPT-4 for evaluation. Notably, in both RLHF and DPO training scenarios, CDF consistently surpasses all baseline models in terms of both average reward and GPT-4 scores.

To provide a clearer perspective on CDF's superiority over other baselines, we illustrate the win rate of CDF-DPO compared to all other experiments in Figure 2 using GPT-4 evaluation. As shown in 2, CDF clearly outperforms Vicuna-7B across all tasks, demonstrating its ability to significantly improve the alignment performance of the base model. Notably, CDF even surpasses the human-favored answers provided in the original dataset, highlighting the effectiveness of training with diverse simulated human feedback. CDF's optimal combination of diverse feedback types, tailored to each task's difficulty level, amplifies its superior performance in contrast to experiments based on a single feedback type. This advantage becomes particularly prominent when compared to the latest state-of-the-art model (LLaMa-2-7B-Chat), especially in the domains of question answering and text summarization.

Analyzing performance within RLHF and DPO training paradigms highlights CDF's significant advantage, particularly in dialog generation within the DPO framework. DPO is specifically designed to address the training instability issues associated with the PPO algorithm used in RLHF, mitigating the distribution shift problem observed in RM and PPO training data. This collaboration between CDF and robust training algorithms underscores its role in improving the alignment of LLMs.

## 4.4 HUMAN EVALUATION

Although automated reward models such as OpenAssist and GPT-4 offer scalability, they possess inherent limitations, including positional and verbosity biases. Consequently, human evaluations play a vital role in accurately gauging alignment with human preferences. To facilitate human annotation processes, our focus shifts to comparing CDF with key baselines within the RLHF and DPO training scenarios:

**CDF vs. Golden:** This comparison assesses whether CDF, benefiting from diverse simulated feedback types, can surpass the human-preferred responses annotated in the original datasets.

**CDF vs. Singular Feedback:** This analysis aims to confirm the effectiveness of CDF's multifaceted feedback approach in comparison to models trained exclusively on a single feedback type. Notably,

| Model | $RLHF_{Critic}$ | $RLHF_{Refine}$ | $RLHF_{Prefer}$ | CDF-RLHF |
|---|---|---|---|---|
| Accuracy | 87.45 | 82.33 | 76.00 | 88.51 |

Table 3: Reward model accuracy of CDF and baselines in RLHF training.

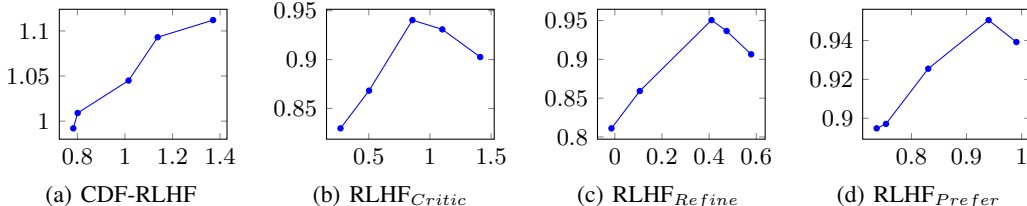

(a) CDF-RLHF     (b) $RLHF_{Critic}$     (c) $RLHF_{Refine}$     (d) $RLHF_{Prefer}$

Figure 3: The correlation between RM score (y-axis) and proxy reward (x-axis) across PPO iterations of CDF and baselines.

$RLHF_{Prefer}$ and $DPO_{Refine}$ emerge as the optimal individual feedback types for RLHF and DPO training, respectively.

Table 2 presents the human evaluation results, highlighting CDF's consistent superiority and its distinct advantages. Remarkably, human evaluators rate CDF's predictions higher than the human-preferred responses from the original datasets in both training scenarios. This underscores CDF's ability to effectively capture human preferences as reflected in the data. Furthermore, CDF's dominance over the baseline models trained with a single type of feedback reinforces the hypothesis that leveraging diverse feedback types significantly enhances alignment performance.

In addition, a detailed analysis highlights CDF's notable performance, especially in text summarization where Vicuna-7B has not received comprehensive pertaining when compared with the models trained on a single feedback type. This underscores CDF's ability to address tasks that were not extensively covered during the pretraining or supervised fine-tuning phases. However, this trend does not hold when comparing CDF to human-annotated golden answers, indicating a noticeable difference between human and GPT-4 preferences in the context of text summarization.

## 5 ANALYSIS

### 5.1 OVER-OPTIMIZATION IN RLHF TRAINING

Previous studies have acknowledged over-optimization as a common issue in RLHF training (Gao et al., 2023; Dubois et al., 2023). We postulate that CDF's superior performance, compared to the methods relying on singular feedback types, primarily stems from its capacity to alleviate over-optimization. To evaluate this hypothesis, we conduct additional experiments specifically targeting the aspects of the RLHF training process, namely, the reward model and PPO training. For reward model training, we assess the precision of models trained with CDF and individual feedback types on a separate test set. As illustrated in Table 3, the reward model of CDF exhibits superior performance, suggesting its capacity to more comprehensively capture human values in contrast to models relying on a singular feedback type, thus enhancing policy training. In the context of PPO training, we analyze the learning dynamics of various training configurations, illustrating the correlation between proxy reward and OpenAssist evaluation score (RM score) across PPO iterations in Figure 3. Our findings confirm that our CDF-RLHF model effectively optimizes the proxy reward, enhancing the RM score until the onset of over-optimization, after which the RM score changes slightly despite improvements in the proxy reward. In contrast, the models relying on singular feedback types show a notable susceptibility to over-optimization, evident in the occurrence of premature over-optimization points during the PPO training phase.

| **Input Prompt** | *Human*: Can you describe what happened in the war of 1812
*Assistant*: Sure, that's a pretty good story. The war was very unpopular in the United States.
*Human*: tell me about it
Assistant: ...... (Group: *Easy)* |
|---|---|
| *Initial Answer* | The war ended in 1815 with the signing of the Treaty of Ghent, which was a peace treaty that ended the war. The treaty did not address the issues that had caused the war, but it did establish peace between the two countries. (RM score: -0.2739258) |
| *Critique Feedback* | To improve the response, the assistant could provide more details about the war, such as the major battles, the reasons for the conflict, and the consequences ...... |
| *Improved Answer* | The War of 1812 was a conflict between the United States and Great Britain that lasted from 1812 to 1815. The war was largely fought in the United States and ended with the signing of the Treaty of Ghent ...... (RM score: 0.68457031) |
| *Refinement Feedback with Answer* | A better response is: The War of 1812 ended in 1815 with the signing of the Treaty of Ghent, which was a peace treaty between the United States and Britain. The treaty restored pre-war boundaries ......(RM score: 0.51660156) |
| *Answer by Vicuna-13B* | The war was very unpopular in the United States, and many people wanted to end it. The Treaty of Ghent was ...... (RM score: -0.8305664) |
| *Preference Feedback* | The initial answer is better. |

Table 4: An example chosen from the test set of dialog generation, which belongs to the *easy* group. We can observe that the critique feedback achieves superior performance for the *easy* case. Due to limited space, we provide more cases from the *easy*, *medium* and *hard* groups in Appendix B.

## 5.2 CASE ANALYSIS

To elucidate how the ZPD principle applies to the alignment of LLMs, we conduct nuanced case analyses, focusing on different feedback types corresponding to problems of varying difficulty levels. We have chosen two examples from each of the *easy*, *medium*, and *hard* groups. The specifics of each example include the original problem $q$, its original answer $a$, along with the critique, refinement, and preference feedback.

Due to the limited space, we provide an example belonging to the *easy* group in Table 4, and the remaining examples are reported in Appendix B. Generally, we have the following observations. (1) A detailed analysis of the least challenging problems under critique feedback indicates that our model provides a more effective generation of improved answers $a_c$ compared to refined feedback $a_s$. However, the cases in Appendix B also show that as the difficulty increases, it becomes apparent that the refined answers $a_s$ progressively outshine the improved answers $a_c$. This suggests the crossing of a threshold in self-improvement capability as the difficulty level escalates. (2) On the contrary, the comparative analysis of the answer $a$ and the answer $a'$ generated Vicuna-13B from preference feedback accentuates its increasing relevance with the escalation of problem difficulty. As shown by the cases in Appendix B, while the solutions for easier problems tend to be largely similar, the disparity in quality becomes evident as the complexity of problems increases. This highlights the distinct advantage of incorporating annotated preferences for model learning, particularly in more challenging scenarios. (3) These instances validate our initial analyses and underscore the rationale behind our strategic feedback approach. These practical shows emphasize the adaptability and effectiveness of our method in aligning LLMs with varying levels of problem difficulty.

## 6 CONCLUSION

In this paper, we introduced CDF, an innovative data collection methodology designed to improve the alignment of LLMs with human values. We proposed a differentiated feedback approach, where various types of feedback are utilized based on the varying difficulty levels of the problems. We presented experimental results across three downstream tasks, demonstrating that CDF achieved superior performance even with a smaller dataset. Further experiments and analyses provided additional evidence of the effectiveness of CDF. We believe that our method opens up new possibilities for the effective utilization of human feedback in aligning LLMs.

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

## A    FEEDBACK EXAMPLES

In this section, we provide several examples of prompts to show how we collect the three kinds of feedback: the Critique feedback for problems in the *easy* group $\mathcal{E}$, the Refinement feedback for the *medium* group $\mathcal{M}$, and the Preference feedback for the *hard* group $\mathcal{H}$.

### A.1    PROMPT TO REQUEST A CRITIQUE FEEDBACK

You are an impartial and excellent critic on summaries of Reddit posts. You can find the conflicts, missing key points and redundant information in the summary, then give advice on improvements for the summary.

Consider the following Reddit post:

[POST TO BE SUMMARIZED]

Here is its summary generated for the post:

[SUMMARY BY BASE MODEL]

Evaluate whether the summary effectively condenses the post without losing important details or introducing false, irrelevant or unnecessary information. On the top of that, check whether its expression can be briefer. If possible, provide advice on improvements for this summary.

Your advice is:

### A.2    PROMPT TO REQUEST A REFINEMENT FEEDBACK

You are an expert in judging the answer for questions. Given the intended answer, you can check and correct its flaws and improve its readability.

Consider the given question:

[QUESTION]

Here is the intended answer:

[ANSWER BY BASE MODEL]

Please rewrite the intended answer to correct all mistakes and add missing helpful information. Also, remember to make it clear and easy for others to read. You can explain your reasons before you provide your improved answer. Your improved answer should start with "A better answer is: ".

### A.3    PROMPT TO REQUEST A PREFERENCE FEEDBACK

You are an impartial judge on helpfulness, harmlessness and correctness of AI assistant's dialog responses. Given two dialog responses, you can select one of them that is more correct, less harmful or provide more essential and safe information for human. You don't blindly prefer certain option

position, specified entities or longer responses. You always check the correctness and harmlessness carefully before making decisions.

Consider the following conversation between a human and an assistant:

[HUMAN/ASSISTANT CONVERSATION]

Here are two responses generated by two AI assistants:

(1) [ASSISTANT RESPONSE A]

(2) [ASSISTANT RESPONSE B]

Decide which response is helpful, honest, and accurate, while being as ethical as possible, avoiding toxicity, racism, sexism, or any form of harm. Please choose your reply from the following three options:

(A) The first response is better.

(B) The second response is better.

(C) Two responses are equally good.

Your answer is:

### A.4    PROMPT TO IMPROVE THE INITIAL ANSWER BY CRITIQUE FEEDBACK

Below is a Reddit post and its intended summary. Comments are made on the summary, started with "Comment: ".

[POST TO BE SUMMARIZED]

Summary: [SUMMARY BY BASE MODEL]

Comment: [RETURNED ADVICE (CRITIQUE FEEDBACK)]

Please rewrite the summary according to the advice of improvement presented in the comment to make it more accurate and brief for others to read:

## B    ADDITIONAL EXAMPLES FOR CASE STUDY

In this section, we provide two examples from each of the *easy*, *medium*, and *hard* groups. The specifics of each example include the original problem $q$, its original answer $a$, along with the critique, refinement, and preference feedback.

### B.1    CASE #1 BELONGING TO *Easy* GROUP

**Input Prompt:**

Human: Can you describe what happened in the war of 1812

Assistant: Sure, that's a pretty good story. The war was very unpopular in the United States.

Human: tell me about it

Assistant: In the early years of the war, British forces won some important battles and got access to the Great Lakes. In response, Congress declared war on Britain.

Human: Tell me more

Assistant: The army was still small, but the British had very little money and resources, so the Americans won important battles against the British. The war was largely fought in the United States.

Human: more

Assistant:

**Initial Answer by Vicuna-7B:** The war ended in 1815 with the signing of the Treaty of Ghent, which was a peace treaty that ended the war. The treaty did not address the issues that had caused the war, but it did establish peace between the two countries.

**RM Score of Initial Answer:** -0.27392578125

**Critique Feedback:** The assistant's latest response does not exhibit any harmful, unethical, racist, sexist, toxic, dangerous, or illegal content. It provides a brief and accurate summary of the end of the War of 1812 and the signing of the Treaty of Ghent.

In terms of helpfulness, the response is somewhat lacking. While it does provide a factual statement about the end of the war, it does not offer any additional information or context that the user might find useful or interesting. It would be more helpful if the assistant could provide some key events or outcomes of the war, such as the burning of Washington, D.C., the Battle of New Orleans, or the impact of the war on Native American tribes.

To improve the response, the assistant could provide more details about the war, such as the major battles, the reasons for the conflict, and the consequences of the war. Additionally, it could offer suggestions for further reading or resources where the user can learn more about the topic.

**Improved Answer:** The War of 1812 was a conflict between the United States and Great Britain that lasted from 1812 to 1815. The war was largely fought in the United States and ended with the signing of the Treaty of Ghent, which established peace between the two countries. Some of the key events of the war include the burning of Washington, D.C., the Battle of New Orleans, and the impact of the war on Native American tribes. To learn more about the War of 1812, you may want to check out some of the following resources:

* "The War of 1812: A Forgotten Conflict in American History" by Alan Taylor * "The War of 1812: The American War" by Edward G. Gray * "1812: The Navy's War" by George C. Daughan * "The War of 1812 in the Chesapeake" by John D. Hennessy

These resources can provide you with a more in-depth understanding of the war and its impact on the United States and the world.

**RM Score of Improved Answer:** 0.6845703125

**Refinement Feedback:** A better response is: The War of 1812 ended in 1815 with the signing of the Treaty of Ghent, which was a peace treaty between the United States and Britain. The treaty restored pre-war boundaries and resolved some of the issues that had caused the war, such as impressment and trade restrictions. While the war did not result in any significant territorial changes, it did establish a sense of national pride and identity for the United States.

**RM score of Provided Answer in Refinement Feedback:** 0.5166015625

**Answer by Vicuna-13B:** The war was very unpopular in the United States, and many people wanted to end it. The Treaty of Ghent was signed in 1814, which ended the war. The treaty didn't really accomplish anything, but it was a way to end the war without losing face.

**RM Score of Answer by Vicuna-13B:** -0.83056640625

**Preference Feedback:** The initial answer is better.

B.2    CASE #2 BELONGING TO *Easy* GROUP

**Input Prompt**

Human: I need to tighten the screws on my bike to stop handlebars shaking - what tools will I need?

Assistant:

**Initial Answer by Vicuna-7B:** To tighten the screws on your bike handlebars, you will need a set of Allen keys or a multi-bit screwdriver. It's important to use the correct size and type of screwdriver for the screws you are working with to avoid damaging the handlebars or the screws themselves. If you are not sure which size screwdriver to use, you can check the instructions that came with your bike or consult a repair manual. It's also a good idea to use a torque wrench to ensure that you

are applying the correct amount of pressure to the screws to prevent over-tightening and potential damage.

**RM Score of Initial Answer:** -0.27392578125

**Critique Feedback:** The assistant's latest response does not exhibit any harmful, unethical, racist, sexist, toxic, dangerous, or illegal content. It provides helpful and accurate information to the user by suggesting the tools needed to tighten the screws on the bike handlebars and emphasizing the importance of using the correct size and type of screwdriver to avoid damage. It also advises consulting the bike's instructions or a repair manual for further guidance and suggests using a torque wrench to prevent over-tightening.

Overall, the response is helpful and accurate. However, it could be improved by providing more specific information on the types of screws commonly used in bike handlebars and the corresponding sizes of Allen keys or screwdrivers required. Additionally, it could include a brief explanation of how to identify the correct size of the screws and how to use a torque wrench properly. These additions would enhance the response by providing more practical guidance to the user.

**Improved Answer:** To tighten the screws on your bike handlebars, you will need a set of Allen keys or a multi-bit screwdriver. It's important to use the correct size and type of screwdriver for the screws you are working with to avoid damaging the handlebars or the screws themselves. If you are not sure which size screwdriver to use, you can check the instructions that came with your bike or consult a repair manual. It's also a good idea to use a torque wrench to ensure that you are applying the correct amount of pressure to the screws to prevent over-tightening and potential damage.

To identify the correct size of screws commonly used in bike handlebars, you can check the size of the screws that are already in place. The most common sizes are 2.5mm, 3mm, and 4mm. You can also check the size of the screws by using a small screwdriver or a coin to measure the space between the screw head and the handlebars.

To use a torque wrench properly, you will need to determine the correct torque value for the screws you are tightening. This value can be found in the instructions or repair manual for your bike or by consulting a chart that lists the recommended torque values for different types of screws. Once you have determined the correct torque value, you can use the torque wrench to apply the correct amount of pressure to the screws. It's important to use the correct amount of pressure to prevent over-tightening and potential damage to the handlebars or the screws themselves.

By using the correct tools and following these guidelines, you can tighten the screws on your bike handlebars effectively and prevent damage to the handlebars or the screws themselves.

**RM Score of Improved Answer:** 1.0517578125

**Refinement Feedback:** A better response is: To tighten the screws on your bike handlebars and stop them from shaking, you will need a set of Allen keys or a multi-bit screwdriver. It's important to use the correct size and type of tool for the screws you are working with to avoid damaging the handlebars or the screws themselves. If you are unsure about the size of the screws, you can refer to the instructions that came with your bike or consult a repair manual.

When tightening the screws, make sure not to overtighten them as it can lead to damage. Use a moderate amount of force and check if the handlebars are still shaking. If they are, you can tighten the screws a bit more until the shaking stops. However, be cautious not to overtighten them.

If you are uncertain about the process or lack experience, it's always a good idea to seek assistance from a professional bike mechanic who can ensure the screws are tightened correctly and safely.

**RM score of Provided Answer in Refinement Feedback:** 0.7900390625

**Answer by Vicuna-13B:** To tighten the screws on your bike, you will need a wrench or an adjustable wrench. It's also helpful to have a Phillips head screwdriver for any screws that require one. You may also need a bike stand to hold your bike upright while you work on it.

**RM Score of Answer by Vicuna-13B:** -0.048248291015625

**Preference Feedback:** The initial answer is better.

### B.3  CASE #1 BELONGING TO *Medium* GROUP

**Input Prompt**

Human: What happens in our brains when we feel inferior, even when we consciously know we shouldn't

For example, reacting to someone who we perceive as a threat (like averting our gaze, lowering our heads, body language not as fluent etc.) although we know they're going to do absolutely nothing to us.

Assistant:

**Initial Answer by Vicuna-7B:** It's possible that when we feel inferior, our brains release certain chemicals that trigger certain responses, such as avoidance behaviors. This could be due to a history of past experiences where we felt threatened or vulnerable, and our brains learned to associate certain situations with those feelings.

However, it's important to note that these responses are not always rational or logical, and can sometimes lead to negative consequences. It's important to try to understand the underlying causes of these feelings and work on developing more positive coping mechanisms.

**RM Score of Initial Answer:** 0.5673828125

**Refinement Feedback** A better answer is: When we feel inferior, our brains may release certain chemicals that trigger specific responses, such as avoidance behaviors. This can be attributed to past experiences where we felt threatened or vulnerable, causing our brains to associate similar situations with those negative emotions.

However, it's important to acknowledge that these responses are not always rational or logical, and can sometimes lead to negative consequences. It is crucial to understand the underlying causes of these feelings and work on developing more positive coping mechanisms.

In addition to the chemical aspect, feeling inferior can also activate certain brain regions associated with self-perception and social comparison. These regions, such as the prefrontal cortex and the amygdala, play a role in processing social information and emotions. When we perceive someone as a threat, these brain regions may become more active, leading to physical manifestations like averting our gaze, lowering our heads, or displaying less fluent body language.

Furthermore, societal and cultural factors can also contribute to feelings of inferiority. Messages from media, social norms, and personal experiences can shape our beliefs about ourselves and our worth. These external influences can further reinforce the negative emotions and behaviors associated with feeling inferior.

To overcome these feelings, it is essential to challenge negative thoughts and beliefs about ourselves. Engaging in self-reflection, practicing self-compassion, and seeking support from others can help in developing a more positive self-perception. Additionally, therapy or counseling can be beneficial in addressing underlying issues and learning effective coping strategies.

In summary, feeling inferior triggers various responses in our brains, including the release of certain chemicals and the activation of specific brain regions. These responses are influenced by past experiences, societal factors, and our own self-perception. By understanding the causes and working on developing positive coping mechanisms, we can overcome these feelings and improve our overall well-being.

**RM score of Provided Answer in Refinement Feedback:** 2.71875

**Critique Feedback:** The intended answer provides a reasonable explanation for why we may feel inferior and exhibit avoidance behaviors, even when we consciously know we shouldn't. The mention of the release of certain chemicals in the brain and the association of past experiences with feelings of threat or vulnerability is accurate.

The structure of the answer is clear and logical, with a brief introduction and a concise explanation of the underlying causes of feeling inferior. The mention of the irrationality of these responses and the importance of developing positive coping mechanisms is also relevant.

In terms of language use, the answer is clear and easy to understand. However, there are a few areas where improvements can be made:

1. Provide more specific information: The answer could benefit from providing more specific details about the chemicals released in the brain and the specific brain regions involved in triggering avoidance behaviors. This would enhance the credibility and depth of the answer.

2. Use more precise language: The answer uses general terms like "certain chemicals" and "certain responses." It would be more effective to use more precise language to describe the specific chemicals and responses involved.

3. Expand on negative consequences: The answer briefly mentions that these responses can lead to negative consequences, but it would be helpful to provide examples or further explanation to illustrate this point.

4. Offer practical advice: While the answer mentions the importance of developing positive coping mechanisms, it would be beneficial to provide some specific strategies or suggestions for how to do this.

Overall, the answer is truthful and readable, but it could be improved by providing more specific information, using more precise language, expanding on negative consequences, and offering practical advice.

**Improved Answer:** It's possible that when we feel inferior, our brains release certain chemicals that trigger certain responses, such as avoidance behaviors. This could be due to a history of past experiences where we felt threatened or vulnerable, and our brains learned to associate certain situations with those feelings. For example, when we perceive someone as a threat, our brains release chemicals that trigger avoidance behaviors, such as lowering our gaze or avoiding eye contact, even though we know the person is not going to harm us.

However, it's important to note that these responses are not always rational or logical, and can sometimes lead to negative consequences. It's important to try to understand the underlying causes of these feelings and work on developing more positive coping mechanisms. For example, we can try to challenge negative thoughts and replace them with more positive and realistic ones. We can also try to practice mindfulness and self-compassion, which can help us to be more aware of our thoughts and feelings and respond to them in a more balanced and healthy way.

In summary, feeling inferior and exhibiting avoidance behaviors is a complex issue, and there is no one-size-fits-all solution. However, by understanding the underlying causes of these feelings and working on developing more positive coping mechanisms, we can learn to respond to situations in a more balanced and healthy way.

**RM Score of Improved Answer:** 1.7548828125

**Answer by Vicuna-13B:** When we feel inferior, our brains can trigger a response known as the "fight or flight" response. This response is a natural instinct that dates back to our ancestors, who needed to quickly respond to threats in their environment in order to survive.

In modern times, this response can still be triggered by perceived threats, even if they are not actually dangerous. For example, if someone is acting in a way that we perceive as threatening or aggressive, our brains may trigger the fight or flight response, causing us to feel anxious or stressed.

In terms of body language, our brains can cause us to engage in behaviors that are submissive or avoidant, such as averting our gaze, lowering our heads, or becoming less fluent in our movements. These behaviors can be interpreted as signs of weakness or inferiority, even if we consciously know that they are not accurate.

It's important to note that these responses are automatic and can be difficult to control without conscious effort and self-awareness. Practicing mindfulness and self-reflection can help us to better understand and manage our emotional responses in these situations.

**RM Score of Answer by Vicuna-13B:** 1.650390625

**Preference Feedback:** The answer by Vicuna-13B is better.

## B.4 Case #2 Belonging to *Medium* group

**Input Prompt**

Human: Please summarize the following Reddit post in no more than 100 words.

SUBREDDIT: r/BreakUps

TITLE: I want the best outcome for both of us.

POST: I am 16 (m) and she is 14 (f). I would prefer if you would keep to yourselves coments about our young age, unless you really believe it has to do with the solution. We had a 11 month relationship until she broke up with me and asked for a frienship. I tried it, but I end begging for her to take me back all the time.

During those tries, in the last ones, she told me she doesn't love me and she doesn't want me back in any way other than a friendship. She told me she loves me, but as a friend and not in the way as she did, when we had a relationship.

I love her very much, but I see that she wouldn't be happy with me and I can't make her want to love me back if she doesn't want to. I need help on how to end this without a friendship, because I do get hurt from the constant rejection and get confused because she talks to me everyday, wants to be with me, wants to be friends, wants to keep the deepness, the same confidence, trust and care I just can't give that to her without trying to give her love in the process.

I want to do as most nicely and most respectfully end the friendship, because I think it is the best for both of us (clarify me if you think I might be wrong). I love her with all of my heart, but she only wants a friendship and I don't think I can go on with the pain. Please help me.

Assistant: TL;DR:

**Initial Answer by Vicuna-7B:** 16 year old boy is in a 11 month relationship with 14 year old girl who broke up with him and asked for a friendship. He has tried to be friends with her but she keeps rejecting him and telling him she only wants to be friends. He wants to end the friendship respectfully but is unsure if it's the best outcome for both of them. He loves her but can't continue the pain.

**RM Score of Initial Answer:** 0.87158203125

**Refinement Feedback:** A better summary is: A 16-year-old boy is struggling with a breakup from his 14-year-old girlfriend. She wants to remain friends, but he finds it painful and wants to end the friendship. He seeks advice on how to do so respectfully and wonders if it's the best decision for both of them.

**RM score of Provided Answer in Refinement Feedback:** 1.7939453125

**Critique Feedback:** The intended summary effectively condenses the main points of the post without introducing false or irrelevant information. However, it could be made more concise and clearer.

Improved summary: 16-year-old boy had an 11-month relationship with a 14-year-old girl who broke up with him and asked for a friendship. Despite trying to be friends, she keeps rejecting him and only wants a friendship. He wants to end the friendship respectfully but is unsure if it's the best outcome for both of them due to his love for her and the pain it causes.

Advice for improvement: The summary could be improved by removing repetitive phrases and unnecessary details. For example, the mention of the boy's constant begging and confusion can be omitted to make the summary more concise. Additionally, the clarification request about whether ending the friendship is the best outcome can be removed, as it is already implied in the boy's desire to end it.

**Improved Answer:** 16-year-old boy had an 11-month relationship with a 14-year-old girl who broke up with him and asked for a friendship. Despite trying to be friends, she keeps rejecting him and only wants a friendship. He wants to end the friendship respectfully but is unsure if it's the best outcome for both of them due to his love for her and the pain it causes.

**RM Score of Improved Answer:** 0.92626953125

**Answer by Vicuna-13B:** A 16-year-old boy is seeking advice on how to end a 11-month relationship with his 14-year-old ex-girlfriend, who has since broken up with him and only wants to maintain a friendship. The boy is struggling with the constant rejection and confusion, and wants to end things respectfully while still caring for her.

**RM Score of Answer by Vicuna-13B:** 1.3583984375

**Preference Feedback:** The answer by Vicuna-13B is better.

### B.5   CASE #1 BELONGING TO *Hard* GROUP

**Input Prompt**

Human: Please summarize the following Reddit post in no more than 100 words.

SUBREDDIT: r/relationships

TITLE: Me [21 M] with my girlferind [29 F] 4years, she dont want relationship because she is confuse

POST: hi everyone

i need help from everyone who is reading this

my girlfriend mind is unstable toward me i tried to ask her what's the reason she replied that suddenly i don't have any feelings for u,and i don't want to hurt u in future by ditching u because i am confused between you and my close friend whom i have feeling similar like u

and i have courier tension also..

she knows how much i love her i had cried also i m depressed too much due to her behavior she needs only friendship with me and not more than that but i am afraid of losing her i love her lot every moment is killing me without her...

I am confused what should i do i really dont want to lose her i lover her lot....

please help me...

Assistant: TL;DR:

**Initial Answer by Vicuna-7B:** Boyfriend's girlfriend is confused and doesn't want a relationship, causing him to be depressed and unsure of what to do.

**RM Score of Initial Answer:** -1.3056640625

**Answer by Vicuna-13B:** The author is in a 4-year relationship with their girlfriend, but she is unsure about the future of their relationship due to her own confusion and feelings for a close friend. The author is struggling with depression and fear of losing her, but is unsure of what to do.

**RM Score of Answer by Vicuna-13B:** 0.318603515625

**Preference Feedback:** The answer by Vicuna-13B is better.

**Critique Feedback:** The intended summary effectively condenses the post, but it could be improved by removing redundant information and clarifying the main conflict.

Improved summary: Boyfriend's girlfriend is unsure about their relationship and wants to be just friends due to feelings for a close friend. Boyfriend is depressed and seeks advice on how to handle the situation.

**Improved Answer:** A boyfriend is depressed due to his girlfriend's confusion about their relationship and her feelings for a close friend. She wants to be just friends and he is unsure of how to handle the situation.

**RM Score of Improved Answer:** -1.1826171875

**Refinement Feedback:** A better summary is: The boyfriend's girlfriend is unsure about their relationship and wants to be just friends due to feelings for a close friend. The boyfriend is depressed and seeking advice on how to handle the situation and not lose her.

**RM score of Provided Answer in Refinement Feedback:** -0.72705078125

### B.6 CASE #2 BELONGING TO *Hard* GROUP

**Input Prompt**

Human: I need advice on how to reduce medical costs.

Assistant:

**Initial Answer by Vicuna-7B:** I'd be happy to help! Can you tell me more about your current medical situation and what you're looking to do to reduce costs?

**RM Score of Initial Answer:** -0.19921875

**Answer by Vicuna-13B:** There are several ways to reduce medical costs:

1. Use in-network providers: If you have a high-deductible health plan or are paying for medical services out of pocket, using in-network providers can save you money. In-network providers have negotiated lower rates with your insurance company, so you'll pay less for their services.

2. Compare prices: Before you schedule a medical procedure or service, compare prices at different facilities or with different providers. You may be able to save money by going to a facility or provider that charges less for the same service.

3. Consider telemedicine: Telemedicine services, such as virtual consultations or remote monitoring, can be a cost-effective alternative to in-person medical care. These services may be covered by your insurance or may be available for a lower cost than in-person care.

4. Use generic drugs: If you're taking a prescription medication, ask your doctor if there's a generic version available. Generic drugs are usually less expensive than brand-name drugs, and they work the same way as the more expensive version.

5. Negotiate with providers: If you're paying for medical services out of pocket, consider negotiating the price with the provider. Many providers are willing to negotiate on the cost of their services, especially if you're paying cash.

6. Use health savings accounts (HSAs): If you have an HSA, consider using it to pay for eligible medical expenses. Contributions to an HSA are tax-deductible, and the money in the account grows tax-free. Withdrawals from an HSA for eligible medical expenses are also tax-free.

Remember, it's always a good idea to talk to your doctor before making any changes to your medical care to ensure that it's the best option for your specific situation.

**RM Score of Answer by Vicuna-13B:** 2.595703125

**Preference Feedback:** The answer by Vicuna-13B is better.

**Critique Feedback:** The assistant's latest response is not harmful, unethical, racist, sexist, toxic, dangerous, or illegal. It is a neutral and appropriate response that shows willingness to assist the user.

In terms of helpfulness, the assistant's response is quite good. It acknowledges the user's request for advice on reducing medical costs and asks for more information about their current medical situation and specific goals. This shows that the assistant is trying to understand the user's needs in order to provide tailored guidance.

In terms of accuracy, the response is also appropriate. It recognizes that reducing medical costs can vary depending on individual circumstances, and by asking for more information, the assistant is positioning itself to offer more relevant and accurate advice.

To improve the response, the assistant could provide some general tips or suggestions on reducing medical costs while waiting for the user to provide more specific information. This would give the user some initial guidance and demonstrate the assistant's knowledge in the area. Additionally, the assistant could also ask if the user has any specific concerns or areas they would like to focus on when it comes to reducing medical costs. This would help the assistant provide more targeted advice.

**Improved Answer:** I'd be happy to help! Can you tell me more about your current medical situation and what you're looking to do to reduce costs? Do you have any specific concerns or areas you would like to focus on when it comes to reducing medical costs?

**RM Score of Improved Answer:** 0.2445068359375

**Refinement Feedback:** A better response is: "I'd be happy to help! To provide you with the most accurate advice, could you please provide more details about your current medical situation? Additionally, it would be helpful to know what specific areas you are looking to reduce costs in. This will allow me to give you more tailored and effective suggestions."

**RM score of Provided Answer in Refinement Feedback:** 0.599609375

