# OpenReview forum: "Constructive Large Language Model Alignment with Diverse Feedback"
_ICLR.cc/2024/Conference — ICLR 2024 Conference Withdrawn Submission_

### Official Review · Reviewer_wpoj · 2023-10-29

**Soundness:** 3 good
**Presentation:** 3 good
**Contribution:** 3 good
**Rating:** 6
**Confidence:** 3

**Summary:**

The paper proposes to train language models with different forms of feedback according to the difficulty of a problem, as approximated by the perplexity of the output found by greedy search. The authors show that the proposed method improves the performance across a wide range of tasks.

**Strengths:**

- The proposed method is interesting is novel.
- The experiments seem sound and justify the main claims.

**Weaknesses:**

- I am uncertain about the intuition that easy problem benefits the most from critique, medium benefits from demonstration, and difficult benefits the most from preference. Even though the citation (Vygotsky & Cole, 1978) has been provided, it'd be helpful to explain more intuition based on this citation, since it seems quite unintuitive.
- I am uncertain whether perplexity indeed reflects the difficulty of an input question, since sometimes it might reflect how many different ways to express the answer (see questions below) [1].

Based on the concerns above, I am worried that the empirical gain might come from a reason other than appropriate grouping of "difficulty", since neither the motivation to train the LM for each group in the proposed way nor the measurement of difficulty seems convincing to me.

[1] Surface form competition: Why the highest probability answer isn't always right

**Questions:**

- I saw how the methods used the preference data by using RLHF and DPO. How did the paper use the signal of natural language feedback and (critic) and refine?
- Can you provide citations for how "Similar to how humans may express uncertainty when dealing with challenging questions during exams, a high perplexity score typically indicates that the model is also experiencing a hard question with a high degree of uncertainty while generating its answer."? High perplexity could also be a result of having many different ways to express the answer, so that each answer has low probability. e.g., "Yes" = "of course" = "sure" ...
- Can you show some RANDOM example of inputs with different difficulty and check whether they indeed follow humans' intuition of difficulty?

---

### Official Review · Reviewer_Amop · 2023-11-01

**Soundness:** 2 fair
**Presentation:** 3 good
**Contribution:** 2 fair
**Rating:** 3
**Confidence:** 4

**Summary:**

The paper studies LLM alignment via different forms of feedback. The feedback form is determined based on the difficulty level of each input example when attempted by the unaligned model. Given a set of alignment examples, the paper first ranks them according to their perplexity estimated by the unaligned model (Vicuna-7B). Then, the ranked examples are evenly split into three groups: 1) easy (low perplexity) for which a critique type of feedback is elicited, 2) medium (medium perplexity) for which a refinement feedback is elicited, and 3) hard (high perplexity) for which preference feedback is elicited. To elicit critique feedback, an LLM (GPT-3.5-Turbo) is first asked to criticize the initial answer and then asked to revise it accordingly. To elicit refinement feedback, ab LLM is directly asked to refine the initial answer. To elicit preference feedback, another answer is produced via a different LLM (Vicuna-13B) and another LLM (GPT-3.5-Turbo) is asked to decide which of the two answers is better. Out of the three forms of feedback, the paper collects pairs of answers with one of them marked as preferred and uses them to apply straightforward RLHF or DPO. Automatic and human evaluation on question answering, dialog generation and text summarization show that the proposed approach outperforms relying on an individual form feedback.

**Strengths:**

1. The proposed approach shows significantly improved results compared to using an individual form of feedback.

**Weaknesses:**

1. The proposed method does not really have much to do with 'human feedback'. I only view it as a way for automatically collecting preference examples (pairs of outputs with one is preferred over the other) and then everything else in the traditional alignment pipeline is exactly the same. While the paper claims different forms of feedback are used, all forms are converted to preference pairs that are used to train a reward model in RLHF or used directly in DPO. The reward model in RLHF no longer distinguishes examples based on difficulty as claimed.

2. While the paper shows improved end-to-end results when combing the three forms of "feedback", the paper lacks quality assessment of the automatically generated preference pairs.

3. [Minor] the paper needs to provide citations for "the constructivist learning theory" and the three tasks used for data generation and evaluation.

**Questions:**

1. In the preference-style of feedback, why did you use another larger LLM (Vicuna-13B) to generate a different answer?  I expect the answer of the larger model will be preferred.

2. Instead of using perplexity for difficulty categorization, have you tried using task-specific metrics?

3. The paper needs to provide some details on using GPT-4 for evaluation. "rate the generated content on a scale from 1 to 5, where higher scores indicate better alignment with human values" is very vague and it is unclear how you define "human values" to GPT-4. Did you verify that GPT-4 produce ratings as expected?

---

### Official Review · Reviewer_zeGx · 2023-11-06

**Soundness:** 2 fair
**Presentation:** 3 good
**Contribution:** 2 fair
**Rating:** 5
**Confidence:** 4

**Summary:**

The paper proposes an approach to learning from feedback by combining different feedback families. Inspired by Constructivist Learning, the paper divides the given problems into three difficulty levels depending on the model perplexity and then collects different types of feedback for each problem depending on the assigned difficulty. Then the feedback collected is used to construct pairwise preference data, which is then used to train a reward model as done in the standard RLHF setting. The paper shows improvements as a result of the proposed method over three different tasks.

**Strengths:**

- Combining different families of natural language feedback is an interesting direction.
- Leveraging concepts from human learning theory and using them to improve LLM learning from feedback is unique and novel.
- The results show the proposed CDF method to outperform the baselines.

**Weaknesses:**

- The connection with the Zone of Proximal Development principle is not intuitive: For example, I fail to see why learning from preference should be used with hard or "unsolvable without assistance" problems as the authors suggest. If a problem is completely unsolvable by a student, showing the student two solutions where one is better than the other is unlikely to be helpful.

- The feedback collection process is LM-specific: As the difficulty of the problems is determined using perplexity, one would need to repeat the whole process for each new LM since different LMs will result in different easy-medium-hard groupings, making the approach impractical and expensive to use.

- Many design choices seem unjustified. For example, they are using perplexity to measure difficulty (see questions below) and groping problems into three groups. Why not four, for example, or just two: easy and hard? Also, why use another model (Vicuna-13B) to generate alternative answers? for the "prefer" feedback. Why not sample multiple answers as done in the standard RLHF settings?

- The main novelty of the paper is in constructing the feedback data described in section 3.3. However, by converting all feedback into comparison format, CDF is reduced back to the standard pairwise ranking setting.

- In the intro, the authors criticize the data inefficiency of standard RLHF methods. However, I fail to see how CDF is supposed to be more data efficient; Actually, the proposed feedback collection process is likely to be more expensive.

**Questions:**

- The typical RLHF pipeline involves doing supervised fine-tuning (SFT) first before collecting the pairwise preferences. Why do you omit that step and sample directly from the unaligned Vicuna model? I am worried that doing SFT on the policy model before the preference training could affect your results.

- Why is the perplexity of the answer expected to represent the difficulty of a problem as opposed to the nature of the question itself? What if the model is not well calibrated to being with?

---

### Official Review · Reviewer_1jjk · 2023-11-06

**Soundness:** 3 good
**Presentation:** 4 excellent
**Contribution:** 3 good
**Rating:** 5
**Confidence:** 4

**Summary:**

This work proposes to obtain preference dataset via three kinds of methods: providing critique from GPT3.5 and then prompt the base model to generate the revised response, directly providing revision from GPT3.5, and using GPT3.5 to select one from two responses from two models. This work shows that the diverse preference data sources can bring benefits for both reward model and following PPO or DPO methods.

**Strengths:**

1. This work has done comprehensive experiments to validate the effectiveness of the constructed diverse preference data, such as the RM training, PPO and DPO. And the results look impressive.

**Weaknesses:**

1. I am not sure what is the benefit of grouping the prompts into three difficulty levels. This work has been emphasizing on the importance of obtaining the preference data from diverse sources so as to reduce the over-optimization, but then a natural question can be: what about directly collecting all three sources of preference data for all prompts in the dataset without doing the grouping? Could it be even better? One baseline of such a case should be provided to validate the necessity of grouping.
2. From Table 1, it looks like that the QA and Dialogue related benchmark test sets show very small improvement for both the RM and GPT4 eval scores, however, the summary test set show substantial improvement between the proposed method and all baselines including those ablation baselines. This looks very weird. Why could this method present significant improvement over only one dataset among all three? A detailed analysis needs to be presented.
3. In Table 3, how is the "separate test set" constructed here for testing the RM? I see that the the RLHF_{critic} shows the highest accuracy among the three baselines, which is contradictory to my intuition: in this baseline, Vicuna-7b is used to provide the revision, which should be worse than the revision provided by GPT3.5 in the RLHF_{refine} baseline. In this case, why could RLHF_{critic} show better RM performance? The details of the test set here need to be revealed for us to understand the reason.

**Questions:**

1. In Figure 3, what is the "OpenAssist evaluation score" in the y-axis?